# Combined Release of Antiseptic and Antibiotic Drugs from Visible Light Polymerized Biodegradable Nanocomposite Hydrogels for Periodontitis Treatment

**DOI:** 10.3390/pharmaceutics14050957

**Published:** 2022-04-28

**Authors:** Jozsef Bako, Ferenc Toth, Jozsef Gall, Renato Kovacs, Attila Csík, Istvan Varga, Anton Sculean, Romana Zelko, Csaba Hegedus

**Affiliations:** 1Department of Biomaterials and Prosthetic Dentistry, Faculty of Dentistry, University of Debrecen, 4028 Debrecen, Hungary; bako.jozsef@dental.unideb.hu (J.B.); ferenc.toth@dental.unideb.hu (F.T.); 2Department of Applied Mathematics and Probability Theory, Faculty of Informatics, University of Debrecen, 4032 Debrecen, Hungary; gall.jozsef@inf.unideb.hu; 3Department of Medical Microbiology, Faculty of Medicine, University of Debrecen, 4032 Debrecen, Hungary; kovacs.renato@med.unideb.hu; 4Materials Science Laboratory, Institute for Nuclear Research, 4001 Debrecen, Hungary; csik.attila@atomki.hu; 5Department of Periodontology, Faculty of Dentistry, University of Debrecen, 4028 Debrecen, Hungary; varga.istvan@dental.unideb.hu; 6Department of Periodontology, School of Dental Medicine, University of Bern, 3010 Bern, Switzerland; anton.sculean@zmk.unibe.ch; 7University Pharmacy Department of Pharmacy Administration, Faculty of Pharmaceutical Sciences, Semmelweis University, 1085 Budapest, Hungary; zelko.romana@pharma.semmelweis-univ.hu

**Keywords:** photopolymerization, combined drug delivery, nanocomposite, hydrogel, chlorhexidine, metronidazole

## Abstract

The in situ application of the combination of different types of drugs revolutionized the area of periodontal therapy. The purpose of this study was to develop nanocomposite hydrogel (NCHG) as a pH-sensitive drug delivery system. To achieve local applicability of the NCHG in dental practice, routinely used blue-light photopolymerization was chosen for preparation. The setting time was 60 s, which resulted in stable hydrogel structures. Universal Britton–Robinson buffer solutions were used to investigate the effect of pH in the range 4–12 on the release of drugs that can be used in the periodontal pocket. Metronidazole was released from the NCHGs within 12 h, but chlorhexidine showed a much longer elution time with strong pH dependence, which lasted more than 7 days as it was corroborated by the bactericidal effect. The biocompatibility of the NCHGs was proven by Alamar-blue test and the effectiveness of drug release in the acidic medium was also demonstrated. This fast photo-polymerizable NCHG can help to establish a locally applicable combined drug delivery system which can be loaded with the required amount of medicines and can reduce the side effects of the systemic use of drugs that have to be used in high doses to reach an ideal concentration locally.

## 1. Introduction

Periodontitis is defined as an inflammatory disease of the periodontium that affects about 10–15% of the world population [1]. Combined drug delivery systems are more frequently investigated in the medical field, thanks to the success of different antibiotic and anti-cancer therapies. The enhancement of the effectiveness and the possibility of the decrease of the applicable dose of drugs are desirable properties and appropriate directions for the developments. The not well-established usage of antibiotics can lead to the progression of resistance. The use of two or three different antibiotics in a combination can be more effective in treatment [2,3] and local application of the selected antibiotics may increase the effectiveness as it can be up to 100-fold higher administered in the therapeutic doses compared with the systemic usage [4,5].

The applicable doses of drugs are greatly reduced when the treatment is performed only in the affected area [6,7]. In the field of dentistry, periodontitis is a chronic inflammatory disease triggered by periodontal pathogenic bacteria which can result in loss of tooth’s supporting bone and fiber apparatus and the formation of periodontal pockets ultimately leading to tooth loss [8,9]. Since deep periodontal pockets ensure an ideal environment for the proliferation of pathogenic microorganisms, the local administration of antibiotics can be an ideal form of treatment because high therapeutic doses can be reached by significantly lower drug quantities, thus leading to fewer side effects [10]. However, in order to enable adequate effectiveness of the drugs, the delivery systems must cover the affected area (i.e., the periodontal pocket), ensuring that the necessary concentration is obtained and maintained until the desired antimicrobial and clinical effects are reached [11].

In dental practice, the application of visible (blue) light photopolymerization is general; it follows that this can be one of the most convenient ways for the construction of locally useable drug delivery devices and can open a gate to the adaptation of this system to 3D printing technology. The polymer system can be filled into space next to the tooth (e.g., into a periodontal pocket) and after the fast polymerization, the stable crosslinked structure can release the embedded drugs during a longer period. The biopolymer-based hydrophilic systems give a wide range of opportunities, for the creation of local applicable systems because of versatile forms, e.g., gels, hydrogels, nanogels, fibers, strips, or chips. The combination of these forms can optimize the necessary properties, from the mechanical to the chemical and physical behavior. The poly-γ-glutamic acid is water-soluble in the aqueous phase in a negatively charged biopolymer. It is a biocompatible, biodegradable, and edible polymer, with various modification possibilities [12]. The methacryloil-group modified (MPGA) polymer form and an earlier created nanoparticle (PGA-MNP) version can work together as a special composite with the possibilities of the release of the different drugs at different rates [13,14]. The surface volume ratio of nanoparticles ensures the chance to adjust the release rate of one of the active components, while another drug can come out from the matrix in an environment-dependent way. The combination of antiseptic and antibiotic drugs can ensure a broad spectrum of the medical effect and helps in faster healing with fewer side effects.

Periodontal inflammations are polymicrobial infections and periodontitis is diagnosed clinically instead of a microbiological investigation. As a result, using adjunctive antibiotics in periodontitis has tended to be empirical. Antibiotics must be used as an adjunct to root surface debridement and mechanical destruction of the biofilm. These adjunctive treatments can enhance the outcomes of mechanical treatment [15]. Systemic antibiotics may be used in patients who have stage 4 grade C periodontitis, multiple periodontal abscesses, and necrotizing gingivitis/periodontitis. Monotherapies and combination therapies are available. The most effective antibiotics are amoxicillin and metronidazole [15]. Topical antimicrobials are introduced to minimize systemic side effects and avoid problems with patient compliance [16]. To complement phase I periodontal therapy, there are multiple options of antimicrobial agents that can be locally delivered, such as metronidazole, chlorhexidine, minocycline, doxycycline, and tetracycline. However, an important aspect is that all antibiotics must meet the following criteria: the drug must easily reach the targeted area and remain at an effective concentration; furthermore, it should last for a long period of time [17].

The different stimuli-responsive drug delivery systems can ensure the benefit of the natural control of the inflammatory processes when pH is decreased. Many periodontal systems aim to use the advantage of this possibility and therefore, different polymeric systems and mesoporous silica-based drug releasing materials were created and investigated in the last decade. These materials and combinations show better control possibilities and more flexible availability of the drugs directly in time at the place of application [18,19].

Metronidazole (Metr) is a well-known drug in the treatment of periodontal diseases. It is available in different forms and some developments can be found which investigated effectiveness of metronidazole alone or as a part of the delivery system [20,21]. Chlorhexidine (CHX) is a generally applied antiseptic agent that can be used as a “gold standard” during investigations of new drugs or delivery systems. The effectiveness and applicability of this drug are well known in the oral region because of a wide range of antimicrobial effects and adhesive properties [22,23]. In our earlier study, we investigated the individual releases of different antibiotic drugs from the hydro- and the nano-gels, but the combination of the different forms of the polymers and the different drugs can give more specialized treatment possibilities in general, and particularly in the field of dentistry. The aim of this work was to demonstrate an MPGA/PGA-MNP created blue-light photo-polymerizable nanocomposite hydrogel (NCHG), and the investigations of the release profiles of the CHX and Metr drugs on different pH in a Universal Britton–Robinson buffer solution.

## 2. Materials and Methods

### 2.1. Modifications of PGA

The poly-γ-glutamic acid (PGA, MW = 1.2 × 10^6^ Da, from GPC) was purchased from Nanjing Saitaisi Biotechnology Co. (Nanjing, China). Water-soluble 1-[3-(dimethylamino) propyl]-3-ethyl carbodiimide hydrochloride (EDC) (Carbosyth Limited, Compton, Berkshire, UK), 2,2′-(Ethylenedioxy)bis(ethylamine) (98%) (EDA) (Sigma-Aldrich, St. Louise, MO, USA) as a crosslinker and 2-aminoethyl methacrylate hydrochloride (95%) (AEM) (Polysciences Europe GmbH, Hirschberg, Germany) as a methacrylating agent were applied. The methacrylated-poly-γ-glutamic acid (MPGA) polymer was produced as we described earlier [14]. Briefly, PGA was activated by EDC, and methacryloyl-groups were created by AEM. Another component of NCHG, the methacrylated-poly-γ-glutamic acid nanoparticles (PGA-MNP) were made in a two steps reaction [13]. Firstly, nanoparticles were formed. In this step, EDC was used for the increase of the reactivity of PGA, and EDA was used for the creation of crosslinked polymer which formed NPs. In the second step, the remaining carboxyl-groups were activated by a new portion of EDC, and the created PGA-MNPs were modified by the methacryloyl-group used by AEM.

### 2.2. Characterization of the Methacrylation Reaction

^1^H NMR spectra of the modified PGAs were recorded by Proton Nuclear Magnetic Resonance Spectroscopy (^1^H NMR) on a Bruker 200SY NMR spectrometer (200 MHz) instrument. The samples were dissolved in deuterated water (D_2_O) and the chemical shifts were represented in parts per million (ppm) based on the signal of sodium 3-(trimethylsilyl)-propionate-d_4_ as a reference.

### 2.3. Characterization of the PGA-MNPs

The shape of the nano-sized objects were investigated with a dual beam scanning electron microscopy type Thermo Fisher Scientific-Scios 2 (FIB-SEM, Waltham, MA, USA) operated in conventional scanning (SEM) and scanning transmission (STEM) imaging modes. The samples were prepared on a conventionally used copper grid covered with carbon layer by dropping of suspension and drying at room temperature. Applying 30 kV accelerating voltage bright-field STEM images were collected to find the nanoscale spheres. The polymer concentration of 1 mg/mL was adjusted, and it was suspended in 1 *w*/*w*% of OsO_4_ (Sigma Aldrich, St. Louise, MO, USA). To demonstrate the PGA-NPs in the MPGA matrix, SEM mode was used. NCHG samples were prepared from the mixture of 1/3 part of preliminary OsO_4_ solution suspended NPs and 2/3 part of matrix MPGA. The photopolymerized NCHG samples were frozen at −70 °C, and freeze-dried in a Scanlaf Coolsafe 55-4 (Labogene ApS, Lynge, Denmark) Freeze Drier under vacuum at −52 °C for 1 days. The lyophilized broken surface was investigated in a scanning electron microscope operated at low accelerating voltage (2 keV). Applying such low energy and short working distance (2 mm) allows us to study surface morphology of insulating samples without coating it with a conductive layer (e.g., gold), which in some cases can modify the original surface morphology.

### 2.4. Synthesis of NCHGs

MPGA and PGA-MNP-based NCHGs were synthesized by free radical-initiated photopolymerization in solutions of antiseptic and antibiotic drugs. First, 22.2 *w*/*w*% of MPGA and 11.1 *w*/*w*% PGA-MNPs were mixed with the solution of Irgacure 2959 (~99%, Sigma-Aldrich, St. Louise, MO, USA) as a photoinitiator (8 *n*/*n*% calculated for methacryloyl-group), metronidazole (Metr) (98%, Sigma-Aldrich, St. Louise, MO, USA), and chlorhexidine digluconate solution (CHX) (20%, Sigma-Aldrich, St. Louise, MO, USA) as active substances. The concentrations of the drugs were adjusted to 25 and 50 mg/g for Metr and CHX, respectively. The photo-curing of the NCHG samples was performed by a Bluephase 20i (Ivoclar Vivadent AG, Schaan, Austria) dental polymerization unit (hand lamp). The samples were made in a 2 mm depth and 5 mm diameter cylindrical Teflon mold. The reaction times were 60 s with 2000 mW/cm^2^.

### 2.5. Characterization of the MPGA/PGA-MNP NCHG

#### 2.5.1. Mechanical Investigations

The mechanical properties of the NCHGs were investigated with an INSTRON 5544 Universal Mechanical Analyzer (Instron Inc., Norwood, MA, USA). The polymerizability and the effects of the loaded drugs for the main physical parameters were studied with the comparison of the compression test results of the drug-containing systems with the NCHGs without active ingredients. The studies were performed with a full-scale load range at 0.1 kN and crosshead speed at 2 mm/min. The cylindrical hydrogel samples had a diameter of 5 mm and a specimen length of 2 mm.

#### 2.5.2. Swelling Properties

The swelling parameters of the NCHGs were observed by gravimetric analysis. It was carried out by immersion of the samples in adjusted Universal Britton–Robinson buffer solutions (UB-RBS) (pH 4–12). The buffer solutions were individually produced from boric acid 99.5% (Reanal Ltd., Budapest, Hungary), phosphoric acid 85% (VWR Chemicals, Fontenay-sous-Bois, EC), and glacial acetic acid (VWR Chemicals, Fontenay-sous-Bois, France), and were adjusted by NaOH solution (0.2 M) (VWR Chemicals, Leuven, Belgium). The samples were removed from the media and wiped cautiously with bolting paper to eliminate the excess wetness from the surface. The measuring period was 168 h. The weight swelling percentage (Wp) for each sample was calculated as:Wp = (Ws − Wo)/Wo × 100;
where Ws is the weight of the swollen gel and Wo is the original weight of the gel after polymerization.

#### 2.5.3. Study of Drug Release Properties on Different pH

The NCHGs were prepared for release studies with 50 mg/g CHX and 25 mg/g Metr content. The main purpose of these experiments was to examine the release rates of the drugs from the stuffed NCHGs. The samples contained together the drugs in the NCHGs which were immersed in an adjusted UB-RBS (1 mL) (pH 4–12) and subjected to continuous stirring on Heidolph Unimax 1010 plate shaker (Heidolph Instruments, Schwabach, Germany) (100 RPM). At predetermined periods the entire amount of medium (1 mL) was changed, and the concentrations of the drugs were measured by HPLC. A Dionex Ultimate 3000 (Dionex Softron GmbH, Germering, Germany) instrument with Hypersil Gold CN (3 μm) column (Thermo Scientific, Waltham, MA, USA) was used and the absorbances were determined at 258 nm for CHX and 318 nm for Metr. The mobile phase was 70 *w*/*w*% saline-solution (Fresenius Kabi GmbH, Bad Homburg, Germany) with 0.2 *w*/*w*% formic acid (Sigma Aldrich, USA) and 30 *w/w*% acetonitrile (VWR Chemicals, Fontenay-sous-Bois, France), the flow rate was 0.6 mL/min. The removed liquid was replaced by a freshly adjusted buffer solution. The amount of released drugs during the predetermined periods and the cumulative amount of drugs were expressed as the percentage of the original drug content. The determination of the release of the total amount of drugs happened using the same method, but pH 2 buffer solutions, lipase (from hogpancrease 30.1 U/mg) (Sigma Aldrich, St. Louise, MO, USA), and pronase (from Streptomyces griseus 5.05 U/mg) (Sigma Aldrich, St. Louise, MO, USA) were used additionally.

#### 2.5.4. Cell Viability Assay

Human MG63 cell line (ATCC, Manassas, VA, USA) were cultured in Dulbecco’s Minimum Essential Medium (Sigma-Aldrich, St. Louise, MO, USA) containing 10% fetal bovine serum, 1% antibiotic-antimycotic solution, and 1% GlutaMAX (all from Gibco, Life Technologies Co., Grand Island, NY, USA) at 37 °C and 5% CO_2_.

For Alamar Blue assay, 10^5^ cells/well were placed in a 24-well cell culture plate and were left to attach for 24 h. After attachment media were replaced with fresh medium, MPGA/PGA-MNP NCHGs (2 mm × 5 mm gels were used) were submerged using Millipore 24 Well Millicell hanging cell culture inserts 0.4 µm PET (Millipore Co., Billerica, MA, USA), and incubated at 37 °C in a CO_2_ incubator. MG63 cells grown in the absence of hydrogel samples were used as control. After 1, 3, and 7 days, media were replaced with 10 times diluted Alamar Blue reagent (Invitrogen, Life Technologies Co., Eugene, OR, USA), then after 2 h of incubation at 37 °C and 5% CO_2_, the fluorescence of the samples was measured using a microplate reader (HIDEX Sense, Turku, Finland). Hydrogels were removed before and were placed back after the measurements.

#### 2.5.5. Vitality Staining

The hydrogel samples for microscopic analyses were fixed chemically to a glass surface. First, 13 mm diameter #1.5 circle coverslips (Thermo Scientific, Germany) were treated with a 1:1 solution of 48 *v*/*v*% hydrofluoric acid (VWR International, ECR) and distilled water for 1 min, and after cleaning (twice in distilled water and once in acetone), they were modified with silane molecule (Ultradent^®^ Silane, Ultradent Products Inc., South Jordan, UT, USA). After air drying, the NCHG was applied as a thin layer and was chemically attached by 60 s of photopolymerization with a Bluephase 20i dental hand lamp. These samples were placed into a 24-well plate and were disinfected for 30 min by UV light.

In a 24-well plate, 10^5^ MG63 cells/well were seeded onto the hydrogel samples, then incubated for 24, 48, and 72 h. Untreated coverslips were used as a negative control. After the incubation period, the cells were co-stained with fluorescein diacetate (FDA) and propidium iodide (PI) (both from Sigma-Aldrich) for 5 min, at room temperature. Pictures were taken with Zeiss AxioVert A1 inverted fluorescence microscope (Zeiss, Göttingen, Germany).

#### 2.5.6. Antibiotic Release Examination in Agar Plates

First, 0.7% agar at 48 °C was inoculated with *E. coli* K12 ER2738 in the mid-log phase, then mixed and poured into Petri dishes containing a layer of solid 1% agar and left to solidify at room temperature. NCHG hydrogel samples with or without Metr and CHX were placed into 5 mm diameter holes that were prepared into the agar plates. Then, 10 μL pH 2 buffer solution was dropped to the Metr/CHX-containing sample to study the acidic effect on the drug release. The 50 µL Metr/CHX solution was used as a positive control in the experiments. The plates were placed at 37 °C and incubated for 4 and 24 h. After the incubation periods, pictures were taken using a Canon EOS 70D (Canon, Tokyo, Japan) camera, and the diameter of the zone of inhibition was used to assess the release properties of the hydrogels.

#### 2.5.7. Time–Kill Experiments

The antibacterial activity of NCHG disks with and without antimicrobial agents was determined against *Fusobacterium nucleatum* ATCC 25586 reference strain in Fastidious Anaerobe broth (CliniChem Ltd., Budapest, Hungary) supplemented with 0.0005 g/L vitamin K and 0.005 g/L hemin (at pH 4 and pH 7) in a final volume of 5 mL inside the anaerobe chamber. The starting inocula were 2–2.5 × 10^5^ CFU/mL. Aliquots of 0.1 mL were removed at 2 and 8 h; furthermore, at 1, 2, 3, 4, 5, 6, and 7 days they were serially diluted 10-fold and plated (4 × 0.03 mL) onto Schadler anaerobe agar plates and incubated at 37 °C inside the anaerobe chamber for three days. All experiments were performed in triplicate. Time–kill curves were prepared from the calculated living cell number using GraphPad Prism 6.05 (GraphPad Software Inc., San Diego, CA, USA).

### 2.6. Statistical Analysis

All data are shown as mean ± standard deviation (SD). Statistical analysis of viability tests was carried out using the Student’s *t*-test to determine the statistical significance of differences between of experimental groups. *p* < 0.05 was used to determine significance. GraphPad Prism v8 (GraphPad Software Inc., San Diego, CA, USA) was used for the investigations.

In the mechanical data, the comparison of the means of the control and NCHG measures was performed by the independent sample *t*-test or the Welch *t*-test, depending on the equality of variances. The latter condition was checked by Levene’s F test. For all cases, furthermore, a non-parametric counterpart of the above tests, the Mann–Whitney, was also run. The same tests were run for the comparison of the means of NCHG and NCHG with +pH2 medium samples groups regarding the antibacterial effect investigation.

To analyze the relationship between swelling (%) and pH, we fitted standard linear and non-linear regression models, whereas the cumulative releases were explained by a multiple mixed effect regression model, for which the logit transform of the original release was used as dependent variable. The statistical calculations were done in IBM SPSS Statistics (Version 27, IBM Corp., Armonk, NY, USA).

## 3. Results

### 3.1. Modifications of PGA

Characterization of the Methacrylated Components

The methacrylation reactions of PGA were successful as the signals in Figure 1 and Figure 2 show. The signals of chemical shifts of methacryloyl-groups were assigned δ = 6.09, 5.70, and 1.88 ppm. The rate of the methacrylation could not reach the theoretical level (50%) but these materials as parts of the composite were able to react in a fast photo-crosslinking reaction within 60 s, and with this number of methacryloyl-groups, it can be swollen in water. The decreased methacrylation rate is unfamiliar with PGA. Zheng et al. found a similar effect in a comparable reaction [24].

The evidence of the success of the crosslinking reaction was performed by the assignments of the signals of crosslinker moiety from EDA δ = 3.23, 3.62, 3.70 ppm (-CH_2_ groups). The methacryloyl and crosslinker signals in Figure 1 and Figure 2 represent that the chemical modifications were done in the backbone of the PGA molecule and photopolymerizable MPGA polymer and PGA-MNPs were created.

The nanoparticle formation of the crosslinked polymer was proven by STEM images in Figure 3A. The frequency distribution of the size of PGA-MNPs shows that approximately 100 nm diameter round shape particles were created (Figure 3C). In Figure 3B, a large number of round shape or spikes-like formations are visible on the broken surface of the NCHG which are in the 100 nm range as the earlier identified PGA-NPs.

### 3.2. NCHG Preparation

The schematic way of the preparation of NCHGs is presented in Figure 4. The total polymer content is 33.3 *w*/*w*%, in which 2/3 part MPGA and 1/3 part PGA-MNPs were mixed with Metr and CHX solutions, respectively. The photo initiator was added to the complete mixture and after homogenization, 60 s photopolymerization occurred on every sample.

### 3.3. Characterization of the MPGA/PGA-MNP NCHGs

#### 3.3.1. Mechanical Investigations

The mechanical properties of the NCHGs show the evidence that the methacryloyl-groups were reactive and the MPGA matrix forming and PGA-MNPs as nano-components were able to form physically stable hydrogels after 60 s exposure of a hand lamp used every day in dental practice. The picture of these gels is shown in Figure 5. These results could provide a promising basis for further developments, so that this system can be applied to new 3D printing technologies with only minor adaptations.

The results of mechanical investigations (Table 1) show that the drug-containing NCHGs are not as strong as the polymer composition without active ingredients. The 0.1094 MPa compressive stress and the 0.2237 MPa Young-modulus next to the 0.5780 mm/mm strain values were particularly promising because these parameters mean that these polymer systems can work properly as drug delivery systems. These parameters mean that this structure can withstand the forces of the soft tissues next to the teeth [13,25]. The polymer composition without drugs shows much higher compressive stress 0.2924 MPa, and Young-modulus 2.3845 MPa values and less elastic properties with the 0.2224 strain, which means that the filling of this system with any other bioactive components could be able to fulfill the requirements of drug delivery systems and might provide useful tools for other biomedical fields such as tissue engineering. In all cases (Young modulus, load, stress, strain), the control group (NCHG without drugs) and the NCHG filled with drugs clearly showed a significant difference of means by the *t*-tests as well as the Mann–Whitney tests (*p* < 0.001). These mechanical parameters show similar or higher values than others on the biomedical field-composed systems that varied from a few tens to around 100 kPa compressive stress [24].

#### 3.3.2. Swelling Properties

The results of the swelling properties showed that the volume of the NCHGs did not change notably, but depending on the pH, the growth was around 5% in acidic or around 20% in the alkaline medium after one week due to the negatively charged polymer backbone (Figure 6). The statistical analysis proved that the linear models gave better fit to the swelling measures (R = 0.810) than the fitted non-linear ones (exponential, power). Note that the choice of the best curve was not our main goal, one could make further selection of non-linear models on a larger data set. Importantly, the swelling shows a significant positive relationship with the pH value (coefficient = 2.112, *p* < 0.001) as was expected. Naturally, the control sample without drugs showed the lowest swelling result, which is substantially different from the weight increasing of drug-containing NCHGs on other pHs. These are promising results, which suggest that in the condition of application—in an acidic medium—the gels can fill the necessary volume, but it will not grow out from the available spaces. These findings were essential because in some of the biopolymer-based systems, especially with the presence of PGA, swelling behavior could be higher, sometimes more than 200 or 300% [26].

#### 3.3.3. Study of Drug Release Properties on Different pH

The release properties of the NCHG have shown that the different pH of the medium affected the release of the different drugs to varying degrees. The Metr was not substantially affected by the pH, and this drug was released from the composite in the early period of the study. Figure 7 shows that this period is around 8–12 h, which means much longer antibiotic present than the usage of a drug solution. In Figure 8a where the cumulative amounts of released Metr was represented as a function of the complete drug content, this leaching out phenomenon is recognized, but this period can be longer in practice next to the tooth in a pocket where only a small amount of sulcus fluid is flowing slowly. This effect can ensure the continuous presence of the antibiotic drug in around half a day which can be essential, and sufficient for the fast decrease of the number of the bacteria such as *Actinobacillus actinomycetemcomitans*, *Porphyromonas gingivalis*, and *Prevotella intermedia* [27,28]. After this period, the effect of CHX can control the reproduction of the rest of the bacteria. Namely, the pH of the medium had a great influence on the release of the CHX, and especially in the case of pH 4, where the concentration of the drug was continuously above the clinical effective concentration [29,30]. The total released drug concentrations showed that the entire amount of Metr was liberated around 8–12 h from the NCHGs. This effect is very important because it can eliminate the high pathogen presence, and this time period is substantially longer than in a case of simple irrigation with the solution of the drug. The release tendency of the CHX shows strong pH dependence, the liberation of CHX increases with the acidity of the media (Figure 7 and Figure 8b). It is noticeable that the acidic pH—in this case the pH 4—showed the most substantial difference compared to all other pH and provided the best releasing results. In a statistical analysis of cumulative release of results Metr and CHX after the logit transform, we could fit on both variables—as dependent ones—linear regression models with random effect (controlling for the different series of experiments), where time (hours) and pH play the role of a significant independent variable in both models. Namely, (i) for the Metr case, time has a positive relationship with release (coefficient = 0.0102, *p* < 0.001), whereas larger pH values go together with smaller release measures (coefficient = −0.1261, *p* = 0.025), everything else being the same; secondly, (ii) for the CHX case, time has a positive relationship with release (coefficient = 0.0106, *p* < 0.001), whereas larger pH values go together with smaller release measures (coefficient = −0.2666, *p* < 0.001). After one week, half of the embedded CHX comes out from the NCHGs, but Figure 9 shows that all of CHX could be available in the appropriate conditions, because at pH 2, almost the entire amount of the drug was released from the NCHGs, and the usage of the enzymes could not significantly alter the process. The photopolymerization reactions of the NCHGs or the absorbability of the CHX do not cause any decomposition of the drug and allow the continuous release of the active ingredient. In this way, we can reach an antiseptic effect locally in a long term, and do not have to count with the side effect of CHX used in higher concentration, e.g., discoloration of teeth or filling, or bitter taste sensation [31]. The presence of CHX can decrease the activity of the metalloproteinases 2, 8, and 9 of *P. gingivalis*, and reduce the adhesion rate so it is effective in an inflammatory reaction [32,33]. This drug should be more frequently used in periodontal therapy, especially when directly applied in the periodontal pockets [34]. Thus, the locally used antibiotic and antiseptic intra-pocket drug delivery system ensure a higher concentration of active content in the gingival crevice fluid, therefore advanced effectiveness and better patient compliance could be achieved [35]. Different locally used drug delivery systems are developed from the strips, fibers, or microparticles, but next to the opportunities, there are deficiencies of the individual types [36,37,38]. The local administrations provide benefits as less drug is needed but the usage of the appropriate dose is often a challenge in a real situation [39]. The injectable systems can give more flexibility in this regard, but the localization of this drug-containing polymer for a necessarily long time can be a remaining difficulty [40,41]. The hydrogels have versatile crosslinking possibilities from the softer physical interactions to the stronger chemical bonding formations, creating any chances to control the stability in time with the presence of drug, but some more sensitive agents cannot be active after harder initiation processes such as UV-light or higher temperature [42]. Recently, the stimuli-responsive systems have targeted bringing closer the possibilities and the aims, e.g., the pH-dependent drug delivery devices can provide one the most appropriate results in an inflammatory reaction, wound healing, or tissue engineering [43,44].

Until today, several studies have shown a change in the release of CHX as a result of pH change, and more articles can be found as examples for the pH-dependent properties of the PGA, but the combination of these systems can provide the base of novel treatment concepts for patients with moderate or severe periodontitis [6,42,45,46]. The combination of drugs to increase the effectiveness and decrease the necessary amount of the applied medicines is a long-standing endeavor. Aspirin and erythropoietin-filled locally used hydrogels were demonstrated for the treatment of periodontal disease, and another recent example studied the combined application of CHX and ibuprofen in intra-pockets administration and proved their clinical relevance [47,48]. Our newly created CHX/Metr-containing NCHG system is novel, and clinically promising, evidenced by the half-day presence of the antibiotics, and the at least one-week-long-lasting antimicrobial environment. The practical applicability is facilitated by the really fast one-minute visible-light polymerization reaction—directly used by a dental-hand lamp blue-light—and the easy administration possibilities.

#### 3.3.4. Cell Viability Assay

MG63 cells were cultured in the presence or absence of CHX and Metr-containing NCHG samples and cell viability was determined by Alamar Blue assay after 1, 3, and 7 days, and media was changed after each measurement.

After 1 day of incubation, the cell viability was slightly but statistically significantly (*p* = 0.0406) reduced in the presence of the CHX and Metr-containing NCHG samples (Figure 10) compared to the control. However, at the following examination days, this reduction disappeared completely and no changes in the viability of the cells cultured in the presence of the hydrogel sample compared to the control were detected. This observation suggests that most of the unreacted monomers which can cause a reduction in the cell viability were released from the hydrogels on the first day and after the addition of fresh medium, it could not reach an effective concentration to cause a further reduction in the viability.

#### 3.3.5. Viability Staining on the Hydrogel Surface

To investigate whether the cells can grow on the surface of the hydrogels, hydrogel layers were attached to glass coverslips which were placed into the wells of 24-well plates and MG63 cells were seeded onto the surface. Glass coverslips without hydrogels served as controls.

The aggregation of the cells was observed after 24 h of seeding (Figure 11 below) on the hydrogel samples, which became more prominent after 48 h and reached almost total confluency at the 72nd h with the emergence of dead cell clusters, probably inside the origin of the cell aggregates. In contrast, the cells on the coverslips showed normal distribution, proliferation, and viability over the examination period (Figure 11 above).

Therefore, we presume that the hydrogel scaffold presented in this study is appropriate to provide a stable and viable environment for the cells surrounding the site of application.

#### 3.3.6. Antibiotic Effect Investigation

On agar plates, the antibiotic effects of NCHG filled with Metr and CHX were compared to the solution of the same amount of drugs and the effect of the pH 2 buffer solution for the release of the drugs was studied (see Table 2 and Figure 12). An antibiotic effect similar to the control was observed at the drug-loaded NCHG, the circle of inhibition was a little bit smaller in the case of NCHG, but this can be caused by the retaining effect of the gel structure. The inhibition zones were grown to 14.67 mm from 12.54 mm at the solution from 4 to 24 h, and the NCHG has shown an increase from 10.78 mm to 12.39 mm. The effect of acidic buffer on the release was more prominent because after 4 h, it has shown a 24.20 mm diameter inhibition circle, which has not changed substantially after 24 h (23.13 mm). Statistically, the results of antibacterial measures concerning the NCHG and NCHG with pH 2 differed in the mean values significantly both for the 4 and the 24 h measures according to the parametric (*t*) and non-parametric (Mann–Whitney) tests (*p* < 0.02 in all cases). These results proved the notable effect of the pH of this combined drug delivery system and represent the more explicit effectiveness of the acidic environment, e.g., in a closed volume, under an inflamed and sore gum.

#### 3.3.7. Time–Kill Experiments

The killing kinetics of NCHG disks against *F. nucleatum* are presented in Figure 13. The NCHG disks with antimicrobial compounds exerted a marked antibacterial effect at pH 4 and 7. In addition, a remarkable bactericidal effect (at least a log decrease of three in CFU number compared to starting inoculum) was observed from 24 h until the end of the time–kill experiments.

## 4. Conclusions

The photo-polymerizable MPGA- and photo-reactive PGA-MNP-created NCHGs show an alternative route of administration of different dental used drugs. This system can withstand the typical forces which can appear next to a tooth in an inflammatory situation. This photo-curable NCHG ensures more than one-week antiseptic effect over a half a day higher concentration of antibiotic drug-eluting in a lower pH, e.g., in a typical inflammatory situation locally. The biocompatible and biodegradable stock materials and the pH-dependent release properties together with the short-term blue light activation resulted in a practically useable and effective way against the bacteria colonies which cause periodontal inflammations. This study presented a novel possibility of the curing of periodontitis in a comfortable, effective, and long-term manner.

## Figures and Tables

**Figure 1 pharmaceutics-14-00957-f001:**
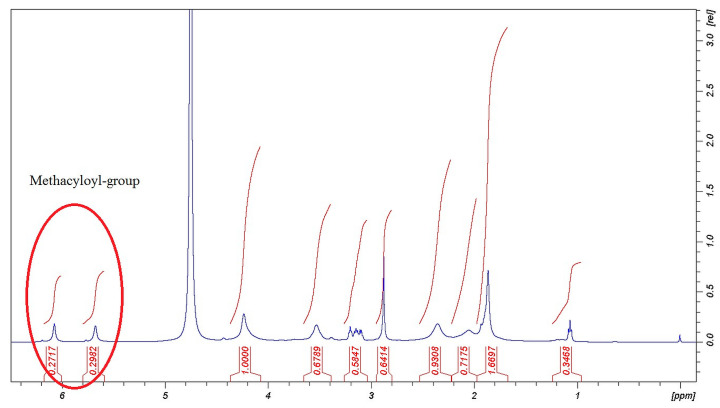
H^1^-NMR spectra of MPGA polymer.

**Figure 2 pharmaceutics-14-00957-f002:**
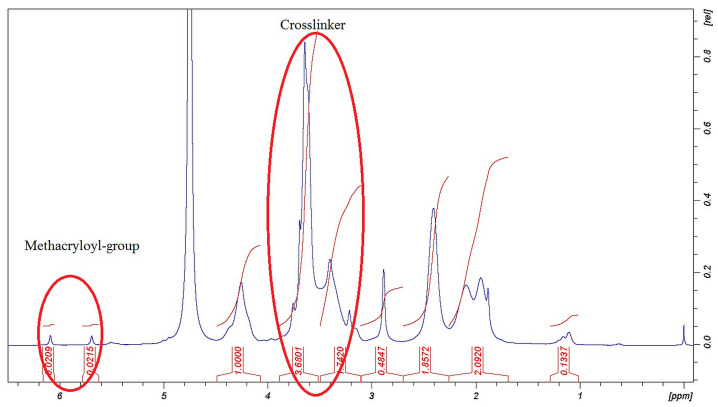
H^1^-NMR spectra of PGA-MNP polymer.

**Figure 3 pharmaceutics-14-00957-f003:**
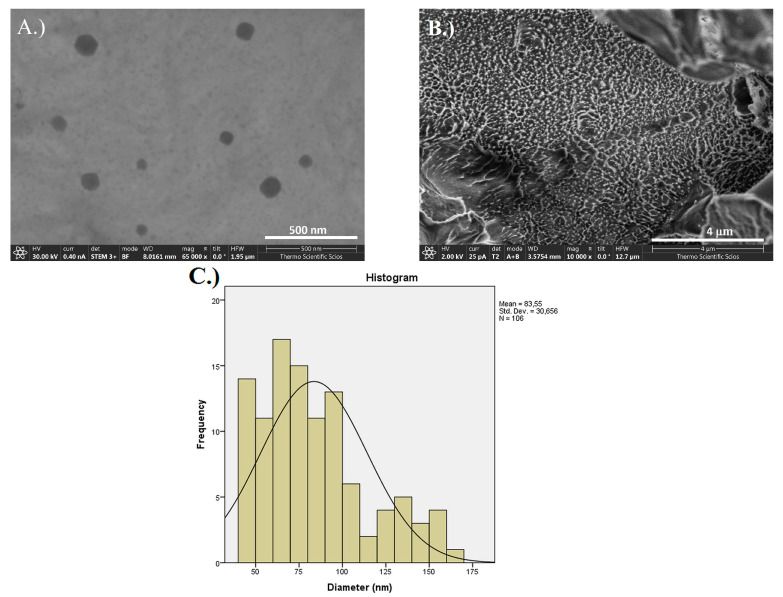
(**A**) STEM images of PGA-MNP polymer, (**B**) SEM image of broken NCHG sample surface, and (**C**) size distribution graph of the nanoparticles.

**Figure 4 pharmaceutics-14-00957-f004:**
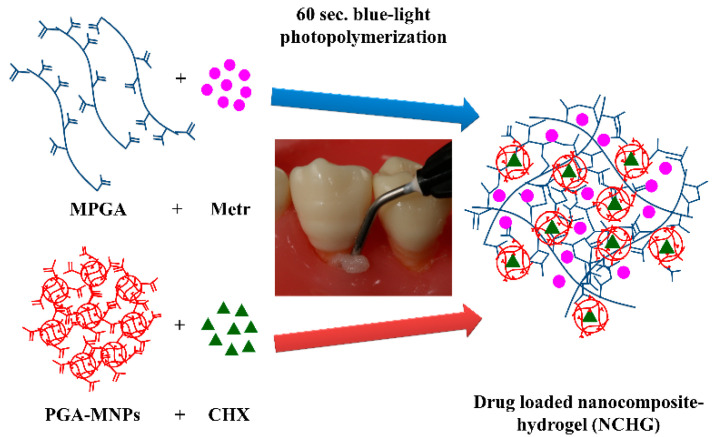
Schematic way of the preparation of MPGA/PGA-MNPs nanocomposite hydrogels.

**Figure 5 pharmaceutics-14-00957-f005:**
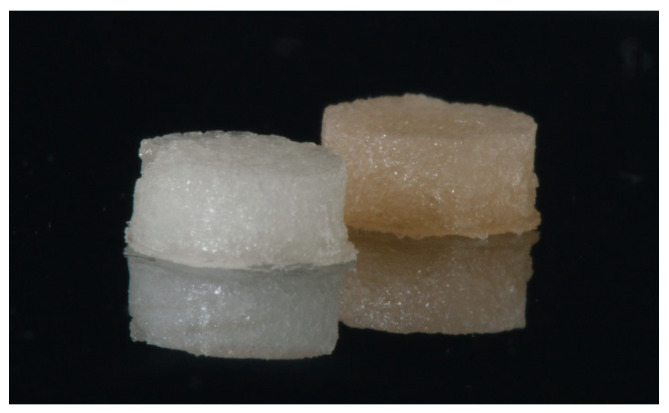
Visible-light photopolymerized NCHGs, the white does not contain drugs, and yellowish shows the CHX and Metr-filled hydrogel.

**Figure 6 pharmaceutics-14-00957-f006:**
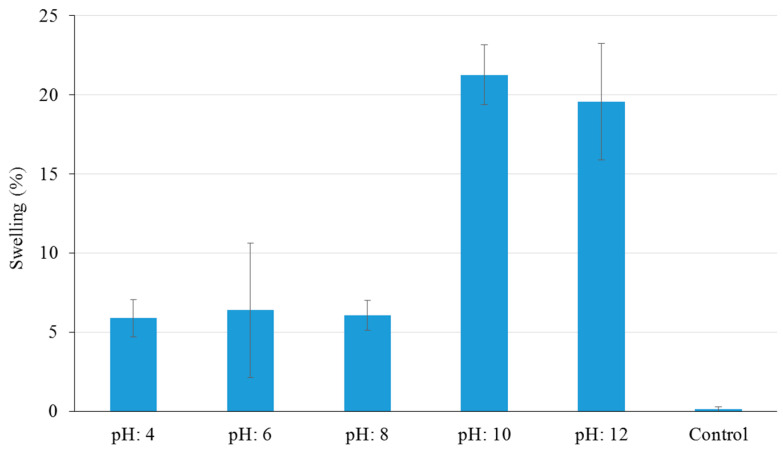
Swelling properties of CHX/Metr-loaded NCHGs in different pH Britton–Robinson buffer solutions after 168 h. Control is the NCHG structure without drugs, error bars represent the standard deviation (SD) of five parallel measurements.

**Figure 7 pharmaceutics-14-00957-f007:**
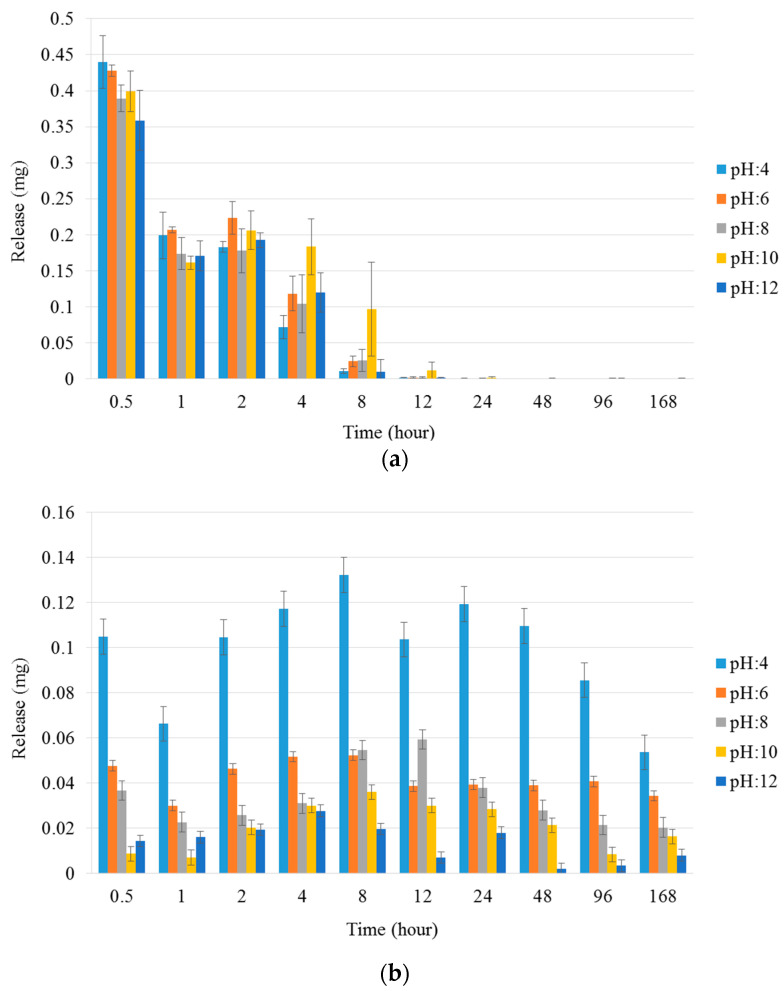
(**a**) Metr and (**b**) CHX release properties of NCHGs in different pH Britton–Robinson buffer solutions over a one week period. Values are expressed as sample means, error bars represent the standard deviation (SD) of three parallel measurements.

**Figure 8 pharmaceutics-14-00957-f008:**
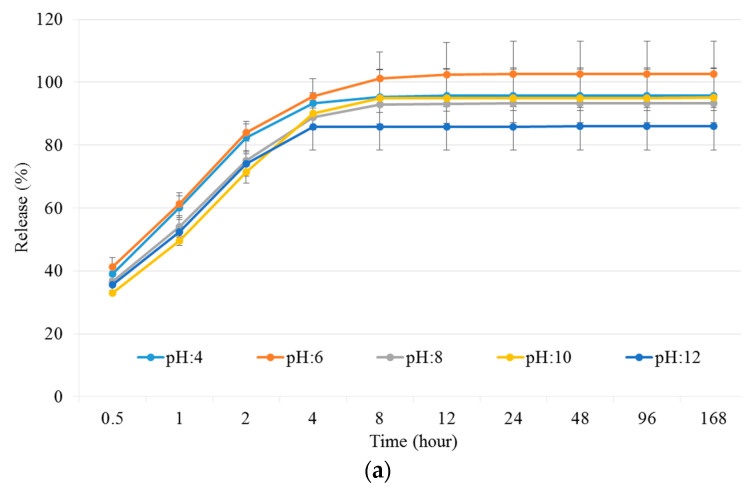
Presentation of cumulative amounts of released drugs in different pHs Britton–Robinson buffer solutions by the accumulation of the released Metr (**a**) and CHX (**b**) in different measuring points and the relations to the entire drug contents. Values are expressed as sample means, error bars represent the standard deviation (SD) of three parallel measurements.

**Figure 9 pharmaceutics-14-00957-f009:**
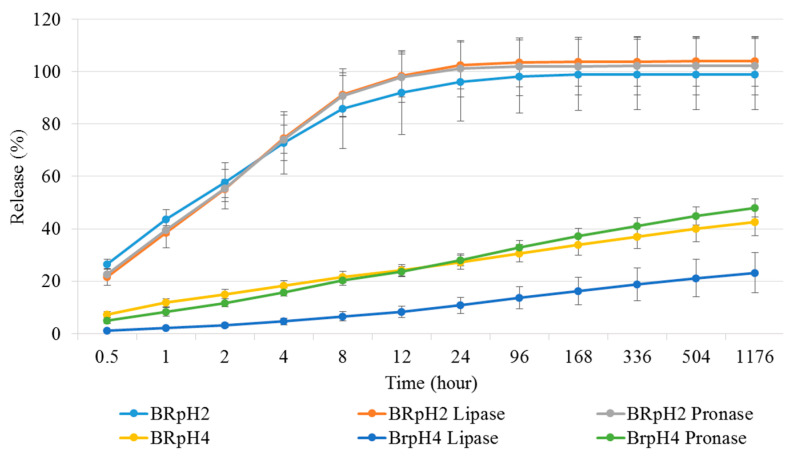
Presentation of cumulative amounts of released drugs in different pHs, and with different enzymes by the accumulation of the released CHX in different measuring points and the relations to the entire drug contents. Values are expressed as sample means, error bars represent the standard deviation (SD) of three parallel measurements.

**Figure 10 pharmaceutics-14-00957-f010:**
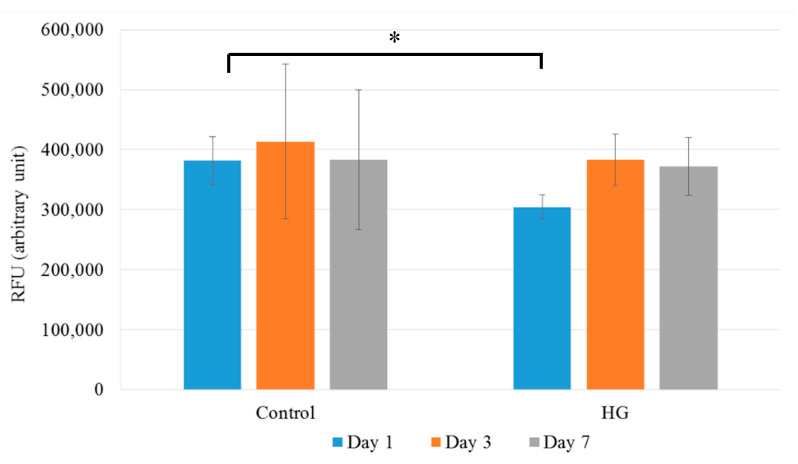
Cell viability assay of MG63 cells: Cells were cultured in the presence (NCHG) or absence (Control) of CHX and Metr containing hydrogel samples for 1, 3, and 7 days. After incubation, cell viability was measured by Alamar blue assay. Values are expressed as sample means, error bars represent the standard deviation (SD) of three parallel measurements, * *p* < 0.05.

**Figure 11 pharmaceutics-14-00957-f011:**
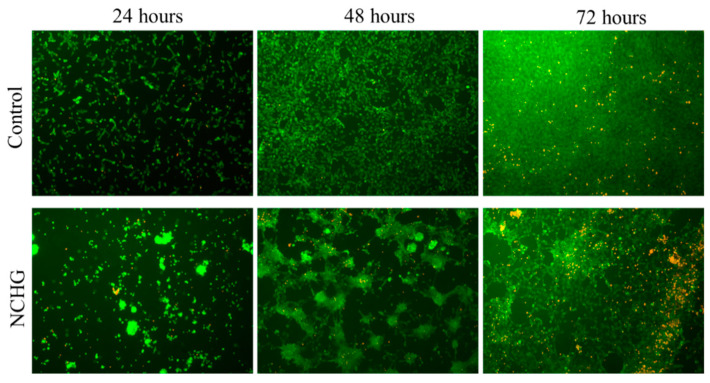
Vitality staining of MG63 cells. The cells were seeded onto hydrogel samples (NCHG) and untreated coverslips (Control) and cultured for 24, 48, and 72 h. After the incubation period, cells on the different surfaces were co-stained by fluorescein diacetate and propidium iodide.

**Figure 12 pharmaceutics-14-00957-f012:**
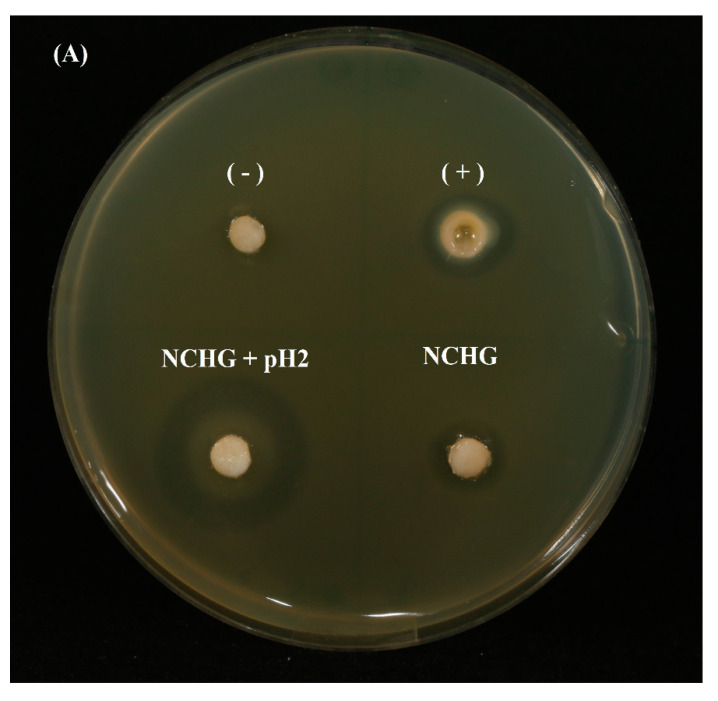
Antibiotic effect of CHX/metronidazole-filled hydrogels: NCHG samples without drugs (−); metronidazole/CHX solution was used as a positive control (+); CHX/metronidazole containing NCHGs were investigated with distilled water (down on the right side), and with pH 2 buffer solution (down and on the left side). (**A**) Picture shows the results at the 4th hour, and the (**B**) image represents the 24 h effects. Samples were placed into agar plates inoculated with *E. coli* K12 ER2738 for the study of antibiotic effect.

**Figure 13 pharmaceutics-14-00957-f013:**
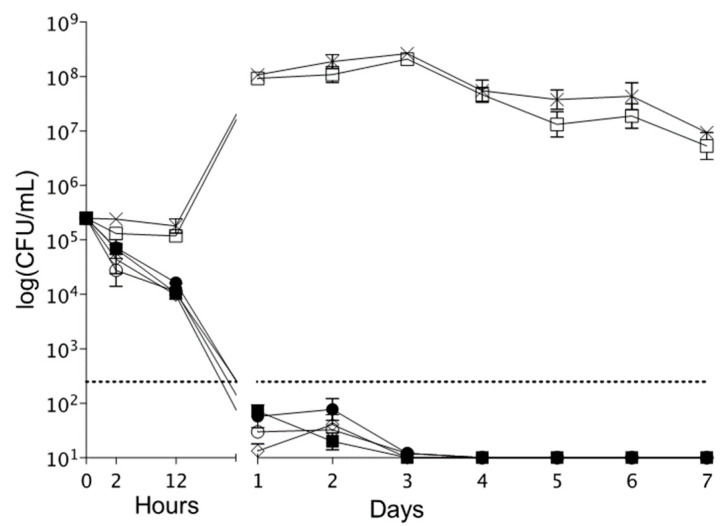
Bactericidal effect of the antibacterial and antiseptic drugs filled NCHGs.

**Table 1 pharmaceutics-14-00957-t001:** Mechanical properties of NCHGs.

n = min. 9	Control (NCHG)		NCHG + CHX + Metr	
	Mean	SD	Mean	SD
Young modulus (MPa)	2.3845	0.3594	0.2237	0.0757
Compressive stress (MPa)	0.2924	0.0178	0.1094	0.0340
Compressive load (N)	4.8806	0.1300	1.8458	0.6466
Compressive strain (mm/mm)	0.2224	0.0251	0.5800	0.0733

**Table 2 pharmaceutics-14-00957-t002:** Diameters of inhibition zones in antibiotic effect investigations.

n = 5	4 h		24 h	
(mm)	Mean	SD	Mean	SD
+	12.5349	0.6613	14.6773	0.9291
NCHG	10.7816	1.3406	12.3976	0.8360
NCHG + pH 2	24.2077	3.1961	23.1351	2.8713

## Data Availability

The data that support the findings of this study are available from the corresponding author, upon reasonable request.

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
