# Peer review of "Combined Release of Antiseptic and Antibiotic Drugs from Visible Light Polymerized Biodegradable Nanocomposite Hydrogels for Periodontitis Treatment"

_pharmaceutics, 2022, doi:10.3390/pharmaceutics14050957_

Round 1

Reviewer 1 Report

The manuscript “Combined release of antiseptic and antibiotic drugs from visible light polymerized biodegradable nanocomposite hydrogels for periodontitis treatment” was focused on the development of methacryloil-group modified (MPGA) polymer and a nanoparticle-based system (PGA-MNP) as a special composite able to release two different drugs (Metronidazole and Chlorhexidine) for treatment of periodontitis. The article is well structured, well written and contains many interesting results and conclusions. However, in my opinion few features might be deepened and require minor revisions.

In particular:

  1. Section 2.4: “Synthesis of NCHGs”. The authors reported that were used “22.2 w/w % of MPGA and 11.1 w/w % PGA-MNPs”. What is the rationale of the use of these MPGA and PGA-MNPs percentages?
  2. Section 2.4: “Synthesis of NCHGs”. Moreover, what is the concentration of metronidazole and chlorhexidine in this preparation?
  3. Section 2.5.4: “Cell viability assay”. Why the authors chosen to study only human MG63 cell line? Probably, in this application, compatibility with gingival cells would be more interesting or equally important.
  4. Section 3.3.2: “Swelling properties”. It is not clear the modality of swelling tests. Have you tested both the bare and drug-containing systems? If yes, in which manner can you distinguish the weight variations due to water uptake from those due to drug release, in the case of drug-containing NCHGs? Moreover, in Lines 378-380 you stated: “Naturally, the control sample without drugs showed the lowest swelling result which is substantially different from the weight increasing of drug-containing NCHGs on other pHs.” Why does the control (i.e., the NCHG structure without drugs) show such a low swelling percentage? Usually, the swelling process is typical of hydrogel systems and it is not linked to the presence of drugs. I would have expected a higher swelling in the empty system and a lower swelling in the drug-containing system, due to drug release.

Reviewer 2 Report

  1. Authors should check affiliation numbering and duplicate entries. If it is desirable to list all the participants' emails, authors could try writing them with the respective name initials at the end of the paragraph instead of repeating the affiliation for each author.

  2. It is well known that saliva acts like a buffer to restore any pH change in mouth to 6.3. This behavior is very fast (minutes). Can the authors comment in the text why did they choose to study the release of the drugs for so long times in varying pH values? Do they expect mouth pH to differentiate for so long in some cases? 
